# The Application of Hyperspectral Imaging Technologies for the Prediction and Measurement of the Moisture Content of Various Agricultural Crops during the Drying Process

**DOI:** 10.3390/molecules28072930

**Published:** 2023-03-24

**Authors:** Ebrahim Taghinezhad, Antoni Szumny, Adam Figiel

**Affiliations:** 1Moghan College of Agriculture and Natural Resources, University of Mohaghegh Ardabili, Ardabil 56199-11367, Iran; 2Department of Food Chemistry and Biocatalysis, Wroclaw University of Environmental and Life Science, CK Norwida 25, 50-375 Wrocław, Poland; 3Institute of Agricultural Engineering, Wroclaw University of Environmental and Life Sciences, Chełmońskiego 37a, 51-630 Wrocław, Poland

**Keywords:** hyperspectral imaging, agricultural products, moisture content, machine learning, modelling

## Abstract

Drying is one of the common procedures in the food processing steps. The moisture content (MC) is also of crucial significance in the evaluation of the drying technique and quality of the final product. However, conventional MC evaluation methods suffer from several drawbacks, such as long processing time, destruction of the sample and the inability to determine the moisture of single grain samples. In this regard, the technology and knowledge of hyperspectral imaging (HSI) were addressed first. Then, the reports on the use of this technology as a rapid, non-destructive, and precise method were explored for the prediction and detection of the MC of crops during their drying process. After spectrometry, researchers have employed various pre-processing and merging data techniques to decrease and eliminate spectral noise. Then, diverse methods such as linear and multiple regressions and machine learning were used to model and predict the MC. Finally, the best wavelength capable of precise estimation of the MC was reported. Investigation of the previous studies revealed that HSI technology could be employed as a valuable technique to precisely control the drying process. Smart dryers are expected to be commercialised and industrialised soon by the development of portable systems capable of an online MC measurement.

## 1. Introduction

The high moisture level of agricultural products will lead to microbial spoilage. Therefore, the chemical reactions and the activity of the microorganisms could be minimised by decreasing the moisture content (MC) of the food products to a specific level, which can prolong the durability of the final product [1]. Dryers are one of the methods of decreasing the moisture level of the samples (Table 1). In this table, the effect of different drying methods on drying time (min) and final and initial moisture content (w.b. (%)) in interval times (min) was shown. The decreasing quality due to the over-drying of agricultural products is one of the most important problems during drying. So, the measurement of MC and the change in the quality of agricultural products during the drying process is necessary as they can affect the drying rate and energy consumption. In other words, over-drying and the rise in the sample temperature during the drying process can adversely affect the quality of the dried samples. Studies have shown that using smart dryers could effectively prevent over-drying the product and further maintain its quality [2].

In this process, the drying condition and quality of the product are continuously and non-destructively controlled [7]. If the MC of the product is not decremented to a specific level, microorganisms will grow, leading to spoilage and moulding. Therefore, the chemical composition of the fruits must be monitored to ensure the quality of the product [8]. Furthermore, a rapid improvement in living standards has increased the public demand for high-quality agricultural products, such that some regulations and laws have been established to guarantee the quality of the products [9]. Mass spectrometry (MS) and high-power liquid chromatography (HPLC) are among the laboratory methods utilised for the quality analysis of the products. Regardless of the accuracy of the results, these methods suffer from several disadvantages, such as being time-consuming and costly [10]. Therefore, the development of a fast, accurate and non-destructive approach for real-time measurement of the moisture content and quality features of agricultural products during the drying process is essential. In this regard, investigating the applicability of the hyperspectral imaging (HIS) technique which can present real-time spectral and spatial data of samples in a non-destructive manner, is of crucial significance [11]. In other words, in addition to detecting the biochemical and physical properties of the samples, the HSI system can even present their corresponding spatial distribution [12]. The advantages of hyperspectral imaging over the traditional methods include minimal sample preparation, non-destructive nature, fast acquisition times and visualising the spatial distribution of numerous chemical compositions simultaneously. In recent years, the hyperspectral imaging technique has been regarded as a smart and promising analytical tool for research, control and industries [13]. The HSI systems are used in various applications such as agriculture, mineralogy, surveillance and target identification, astronomy, chemical imaging, environmental studies and the medical field [14]. The application of HSI in agriculture includes the evaluation of the nutrient status (e.g., nitrogen deficiency [15]), monitoring crop disease [16], estimation of crop biochemical and biophysical characteristics of the crop (e.g., carotenoids, chlorophyll and water contents, LAI, biomass) to understand the physiological status of the vegetation and the prediction of yield and investigation of the properties of the soil (e.g., moisture, organic matter and carbon of the soil) [17,18]. The literature reports food-related studies in which hyperspectral technology was also applied to detect fungal contamination, bruising in apples, faecal contamination, skin tumours in chicken carcasses, grain inspections and so on [19]. The use of HSI technology for measuring the MC of agricultural products during the drying process has been extensively explored [6,20,21,22,23]. The need for fast and careful methods has led to the application of the HSI technology for MC control during the drying process. Moreover, the development of HSI instruments, a new method for data processing, will allow this technology to dominate in the future. This review article attempts to discuss these studies to describe the current state of the application of this technology during the smart drying process of various crops to establish the goals of future investigations.

## 2. HSI Technology

A proper understanding of the HSI technology is essential for some analyses. In this regard, the principles of the HSI technology will be addressed in this section.

The hyperspectral imaging technique is a newly emerging and non-destructive approach that relies on a combination of spectrometry and imaging technologies. According to Sun et al. [24], the mechanism of the HSI technology can be described as follows: At the molecular level, the texture of food products or any biological substance is maintained by several bonds derived from various molecular forces. When electromagnetic waves pass a food sample, their energy will alter due to the bending and stretching vibrations of the chemical bonds such as O-H, N-H and C-H. Spectrometry can offer a precise fingerprint of the food samples based on these alterations in the molecular energy.

At the macroscopic level, electromagnetic waves can be observed as lights that might be reflected, transmitted, or scattered (Figure 1). The choice of these modes depends on the sample type and its components. In the reflection mode, the reflected radiations from the surface of the sample and spectrograph are placed on the side of the light source, and the spectrograph collects the reflected light. This mode can generally detect the external quality of samples, such as damages to the external texture, surface texture, shape and size of the granule samples. In the transmission mode, the spectrograph collects the light passing through the sample, which can offer valuable information about the internal components of the sample. Therefore, this mode is preferably suitable for investigating the internal quality of the samples. This mode is usually employed for the analysis of liquid samples or solid ones such as grains, meats, and dairy. In the scattering mode, which combines the reflexion and transmission modes, the spectrograph and light source are parallel and located on the top of the sample. Compared with the transmission mode, this mode of detection has no thickness limitation in addition to offering more precise information [25]. Therefore, this mode is utilised for large samples such as fruits.

As the absorbed part of the light penetrates the sample, the emission and absorption intensity and wavelength depend on the physical and chemical modes of the sample. Using the technology of HSI, this transmission can be converted into a spectrum and an image (Figure 2). The image obtained is an HSI image showing the chemical composition and physical properties of the food samples [27]. HSI can operate in the wavelength range of 780–2500 nm, which depends on the chemical nature of the food samples [28].

## 3. Components of the HSI Technology

An HSI system includes two main parts: software and hardware. The hardware components of the HSI systems are almost the same. According to Figure 3, the hardware of an HSI system includes a light source that produces light for irradiation of the samples (halogen lamps are often used for this purpose), optical fibres to deliver the light, a spectrograph capable of diffracting the light into various wavelengths, a detector (CCD camera) which converts the optical signals into electrical ones to determine the light intensity, a lens to adjust the focal distance, a conveyer to transfer the samples, and a computer to control the hardware processes. All the software components, such as the software to control the moving scene and receive the HSI images, are installed on the computer. All the components of the HSI system, except for the computer, are placed in a dark box to avoid the effect of the ambient light.

## 4. HSI Acquisition

According to Figure 4, three main methods of point, linear and area scanning can be employed to acquire HSI images. In point scanning (also known as whiskbroom), the spectra of all pixels are individually scanned along the spatial dimensions (x,y) and all the pixels are collected to form a hyperspectral image. Linear scanning (i.e., pushbroom) can be considered an upgrade of point scanning. Once a line of the sample image is obtained along the X dimension, the spectral data of all pixels of this line are recorded. Finally, all the lines form an HSI image. Area scanning (wavelength scanning) differs from the two mentioned methods. Each time an image is obtained in a specific wavelength, it will be repeated in all wavelengths. Finally, all the images of the corresponding wavelengths are overlapped to achieve an HSI image. Among these three methods of HSI acquisitions, the linear scanning method is often utilised to obtain HSI images for quality analysis of agricultural products. This method is capable of one-way and continuous scanning, making it suitable for conveyer systems in food processing lines [30].

## 5. Processes on the Spectra

### 5.1. Principal Component Analysis for Eliminating the Outliers

Principal component analysis (PCA) is one of the multivariable statistical methods with extensive application in the chemical analysis [31,32]. However, due to various reasons such as technical problems, defective data collection and incorrect sampling, some samples might be unsuitable for testing, and they are also called outliers. Therefore, the PCA method is utilised to eliminate these samples before any processing [33].

### 5.2. Pre-Processing

The HSI data are subjected to factors such as light scattering by altering the detector, changes in the sample size, surface roughness and temperature-induced noises, which can affect the calibration accuracy. The data should be thus pre-processed to achieve sustainable, precise and reliable calibration models [8,33]. In other words, as HSI systems are an integration of many different optical and electronic components, pre-processing and calibration of spectral data are necessary for data analysis [19]. Various pre-processing methods have been developed for different goals. However, the choice of the proper pre-processing method is an experimental issue and can be determined by trial and error. Therefore, a specific pre-processing method is not applicable to all prediction models in various samples. So far, smoothing methods (such as Savitzky-Golay, average moving, median filter and Gaussian filter), standard normal variate and multiplicative scatter correction methods have been used to pre-process the spectral data [3]. The smoothing method replaces a specific range of values with a mean value to represent a noisy spectrum as a smooth curve. In addition, pre-processing by SNV and MSC can eliminate the light scattering effect due to irregular shapes and sizes [34].

#### 5.2.1. SNV

SNV is one of the pre-processing methods capable of normalising spectral data. This method can correct the effect of variations in the detector-sample distance and the sample size. In other words, SNV is focused on eliminating the scattered light interference and changes in the path length [35]. In the pre-processing method, all data are often expressed on a similar scale. This pre-processing can be applied for each spectrum through Equations (1) and (2) [36]:(1)$$X_{i,SNV}=\frac{\left(X_{i}-{\bar{X}}_{i}\right)}{SD_{i}}=\frac{\left(X_{i}-\bar{X}\right)}{\sqrt{\frac{\sum_{j=1}^{N}{\left(X_{ij}-{\bar{X}}_{i}\right)}^{2}}{N-1}}}$$
(2)$${\bar{X}}_{i}=\frac{\left(\sum_{j=1}^{N}X_{ij}\right)}{N}$$

#### 5.2.2. MSC

Multiplicative scatter correction (MSC) is one of the well-known normalisations methods capable of correcting the additive (baseline shift) and multiplicative (tilt) effects due to the physical influences on the spectral data. These effects can now be due to the nonuniform distribution throughout the spectrum (the degree of diffusion depends on the wavelength), sample size, and light refractive index [25]. Therefore, MSC aims to separate the chemical absorbance of the light from the light scattering [37]. In this method, the diffusion effects are eliminated by linearisation of the spectra into a mean spectrum (X) according to Equations (3)–(5) [36]:(3)X¯=∑i=1nXijn
(4)$$X_{i}=a_{i}\bar{X}+b_{i}$$
(5)Xi,MSC=Xi−biai

In which:

$\bar{X}$ = spectral matrix with n samples and N wavelengths (i = 1, 2, …, n; j = 1, 2, …, n); a_i_ and b_i_ = simple regression constants of each spectrum against the mean spectrum of the samples, and X_iMSC_ = the corrected spectrum of each sample

This method was utilised in the present research to normalise the spectra.

#### 5.2.3. Smoothing

Smoothing methods (such as Savitzky-Golay, average moving, median filter and Gaussian filter) can decrease the noise in the spectral data. Although these filters can effectively reduce noise, they should be employed with extra care to avoid variations in the important information. Moreover, edge detection or enhancement should not be applied before smoothing. In the Savitzky-Golay algorithm, a polynomial should be matched around several points chosen for smoothing, which can be achieved by a relative minimum square regression method. Smoothing can improve the state of the Vis/NIR spectrum, but it can also result in the loss of important information for the modelling. Despite the extensive investigations to select the optimal point, this selection is made experimentally. An empirical law states that the width of the optimal number of points should not exceed the width required to cover half of the smallest peak of the spectrum [38]. The data include a series of points C_i_,y_j+ii,_) I and j = 1, …, n) in which C_j_ and y_j+ii_ are independent and control variables. According to Equation (6):(6)$$Y_{j}=\sum_{i=\frac{1-m}{2}}^{\frac{m-1}{2}}C_{i}y_{j+i},\frac{m+1}{2}\leq j\leq n-\frac{m-1}{2}$$

Normalisation results in similar data. It also modifies for the change in the conditions.

### 5.3. Calibration Models

Modelling is a monitored classification method used based on the differences between several classes using a specific calibration model to differentiate unknown samples. For instance, Lin et al. (2022) utilised various regression models such as least square support vector machines, nonlinear regression model, partial least squares regression (PLSR) and multiple linear regression between the MC and spectral data of four vegetable samples in various drying intervals [3]. These calibration models were evaluated based on the root mean square error (RMSE) and determination coefficients (R^2^). Generally, a good model should offer the highest R^2^ and lowest RMSE [3]. Furthermore, the residual predictive deviation (RPD) can evaluate the robustness of the model. As Equation (2) showed, RPD is calculated as the ratio between the standard deviation (SD) of the sample reference value to the standard error of prediction of the cross-validated data set. Therefore, researchers suggested that if RPD is >3, the model is robust for screening [39].
(7)$$R^{2}=1-\frac{\sum_{i=1}^{n}{\left(Y_{i}-{\hat{Y}}_{i}\right)}^{2}}{\sum_{i=1}^{n}{\left(Y_{i}-\bar{y}\right)}^{2}}$$
(8)$$RMSE=\sqrt{\overset{n}{\underset{i-1}{\sum }}\frac{{\left(Y_{i}-Y\right)}^{2}}{n}}$$
(9)$$RPD=\frac{SD}{RMSE}$$

### 5.4. Selection of the Optimal Wavelength

HSI encompasses several wavelengths, most of which are not useful to increase the model’s performance as they contain irrelevant information. Some wavelengths may contain similar useful information. Therefore, the modelling process can be simplified by selecting the optimal wavelength and eliminating irrelevant wavelengths to improve the model’s accuracy [18,29]. There are various algorithms to select the optimal wavelength that can be classified into filters, wrappers, and embedded methods [30,40]. Regression coefficient (RC), genetic algorithm (GA), successive projection algorithm (SPA), and competitive adaptive reweighted sampling (CARS). Random frog (RF), spectral derivatives, binary firework algorithm (BFWA), loading coefficient in PCA, ant colony optimisation (ACO) and simulated annealing (SA) are among these methods [11].

Regarding the complexity of the HSI data, several different algorithms should be compared to select the optimal wavelength to determine the best algorithm. Chen et al. (2022) proposed a successive projection algorithm to select the optimal wavelength based on the complete spectral range to predict the chemical composition of the product [8]. Lin et al. (2022) employed RC, CARS [28] and GA algorithms to decrease the number of wavelengths and derive specific wavelengths of the main spectrum [3]. To compare the efficiency of the derived wavelengths, they employed these wavelengths as input data to develop the PLSR, SVM and MLR models.

### 5.5. Development of Models Using Feature Wavelengths

The development of models using the selected wavelengths was performed to simplify the prediction process and improve the calculation speed. One of them was the MLR models [28].

## 6. Comparison of Various Techniques for Prediction of Moisture Content of Agricultural Products

Different techniques are available to determine moisture content during the drying of food, such as computer vision technology (for ginseng roots [41], apple [42]), Near Infrared Reflectance Spectroscopy (for fermented sausages [43], apple [44]), MRI (ham [45]), microwave dielectric spectroscopy (for kiwifruit [46], pork meat [47]). There are many publications on the prediction of MC during various drying methods (e.g., for mango [48], blueberry [49] and apple slices [50]). More evaluation of moisture content by NIRS can be found in [51,52]. However, these techniques have some disadvantages that need to be considered. The computer vision method is not usable for large-scale drying processes in which opaque metal chambers are used. Therefore, improving the scope of use would be a chance for the development of this technology. NIRs method needs calibration based on reference techniques. In addition, the cost of equipment is relatively high, and multiple sensors must be installed to monitor local moisture content or particle size in the dryer.

Moreover, this technique is insensitive to impurities, and only the surface moisture of the material can be determined due to the short wavelength. The use of MRI-enabled visualisation of spatial changes during drying also proves that the MRI method offers valuable information for process optimisation in the commercial processing of different food materials. Still, its application for the online system has a high cost. Microwave dielectric spectroscopy needs more expensive electronic components due to the requirement of applying high frequency, and it must be calibrated separately for different materials. Moreover, it is difficult or even infeasible to attain an appropriate spatial resolution because of the relatively long wavelengths.

In conclusion, the commercial application of microwave dielectric spectroscopy still needs further research [53]. On the other hand, visible and near-infrared (VIS/NIR) spectroscopy was reported as one of the most rapid and appropriate techniques for the prediction and detection of different parameters such as rice and its flour quality [54], tomato pesticide residues [55,56], besides the prediction of MC. But the most important advantages of the application of the HSI technology in the food industry rather than other techniques are as follows [13]:

No requirement for sample preparation; environmentally friendly, non-invasive, and non-destructive method, economical as compared with traditional techniques, recording a spectral volume as compared to spectroscopy (collecting a single spectrum at one spot on a sample), obtaining both qualitative and quantitative properties, reporting the spatial distribution and concentration of the chemical composition of samples, distinguishing samples with similar colours, incomparable for process monitoring and real-time inspection.

### 6.1. Application of the HSI System to Predict the Moisture Content of the Agricultural Products during the Drying Process

As shown in Table 2, studies have shown that numerous articles have addressed the ability of HSI to predict the MC of agricultural products during the drying process. All measurements were carried out through standard experimental methods simultaneously with hyperspectral imaging. The results indicated the ability of the HSI technology in the wavelength range of 350 to 1700 nm [57] to serve as a reference for continuous monitoring of agricultural products during the drying process with the least energy consumption and the highest quality.

Lin et al. (2022) employed the HSI technology to predict the MC of four vegetables (fresh carrot, celery stem, potato, and spinach leaves) during their drying by a vacuum-microwave dryer. They employed a spectrograph made in Finland (ImSpector N17E, Specim, Spectral Imaging Ltd., Oulu, Finland) with a spectral range of 950–1655 nm [3]. They employed the GA-MLR model for the final prediction and declared that this model could offer predictions comparable with other models using complete spectral data. In addition, their model has a simpler implementation procedure in the food industry, and the MLR model is easier and less expensive for industrial real-time systems [58].

In another study, the application of the HSI system and the partial least squares regression model was examined for the measurement and detection of MC and colour of the apple slices during the drying process. The results indicated that the wavelengths of 580, 750 and 970 nm can predict MC with high accuracy [59].

Crighton et al. (2018) used an HSI system manufactured by Specim (Specim Spectral Imaging Ltd., Finland) in the spectral range of 400–1000 nm with the partial least square regression (PLSR) model to predict the moisture content and colour of raw and pre-treated apple slices (Golden Delicious variety). They indicated the capability of the HSI technology of the real-time detection of moisture content and colour of samples during the drying process [4].

Arefi et al. (2021) employed an HSI system, which was a combination of the VisNIR system with a spectrum range of 1010–396 nm and a NIR system with a spectrum range of 1718–937 nm, to predict MC and some qualitative properties of apple slices [11]. Regarding the low sensitivity of camera sensors at the beginning and end of the spectrum, the spectral range below 425 and above 1700 nm was ignored. All the analyses were performed in MATLAB software (version R2020a, MathWorks, Natick, MA, USA). They stated that a combination of spectral data and Gaussian Process Regression (GPR) could accurately predict vitamin content, moisture content and shrinkage of apple samples during drying at wavelengths of 980 and 1450 nm. As both wavelengths were at the same level in terms of prediction accuracy, researchers suggested a wavelength of 980 nm for developing a smart dryer because of the cost-effectiveness of the optical requirements for developing a system based on a wavelength of 980 nm rather than 1450 nm.

Chen et al. (2022) studied the HSI method for online and non-destructive measurement of moisture content and soluble solid content (SSC) of persimmon characteristics during drying [8]. They applied an HSI system made in Belgium with a spectral range of 470 to 900 nm. The distance between the camera lens and the samples was 65 cm with an exposure time of 5 ms. The average spectral value of all the pixels in each desired area of each band was extracted by ENVI Classic software (Research Systems Inc., Boulder, CO, USA). They adopted seven pre-processing methods and regression techniques, including partial least squares regression (PLSR), principal component regression (PCR), least squares support vector regression (LS-SVR) and radial basis function neural network (RBFNN) to analyse the spectral data. Through analysis with MATLAB software (version R2016a, MathWorks, Natick, MA, USA), they revealed that the best regression technique to predict moisture and soluble solids was LS-SVR and PLSR, respectively. These results showed that HSI could be a valuable technique for quantifying chemical compounds in dried persimmon fruits.

A Finnish HSI technology (Spectrometer: Inspector V10E, Spectral imaging, Oulu, Finland, DL-604M, Andor Finland) was employed to evaluate the quality of dried wolfberry fruit under different conditions. SVM, artificial bee colony (ABC), and grey wolf optimiser (GWO) algorithms were utilised to model and optimise. In contrast, Savitsky-Golay (SG) and standard normal variate (SNV) algorithms were used for spectral data pre-processing. The researchers showed that using the HSI technology and model integration (LS-SVM) with SG-SVN pre-processing methods can accurately predict the quality of dried fruit. All the analyses were performed by Unscambler X software and MATLAB R2015 [60].

Seventeen pre-processing methods were adopted for noise elimination of the spectral data to predict the moisture content of potatoes using a Chinese his technology (Zolli Hanguang Co., Ltd., Beijing, China). Four machine learning algorithms, including Extreme Gradient Boosting (XGBoost), Gradient Boosting Categorical Features (CatBoost), Light Gradient Boosting Machine (LightGBM) and Integrated Stacking (Stacking), were compared to predict the moisture content in potatoes. ENVI 4.8 (Exelis Visual Information Solutions, Boulder, CO, USA) software was used for all the analyses. Based on their results, the XGBoost model at the wavelength of 400 nm offered the best prediction of the potato moisture content during the drying process [61].

MC of a carrot was measured during hot air drying using a Finnish HSI technology (V10E PFD, Specim Spectral Imaging Ltd., Finland) in the spectral range of 400 to 1010 nm. The analyses were performed with MATLAB software (versionR2020a MathWorks, Natick, MA, USA), and the results showed the ability of the PLSR model to precisely determine the MC of carrots [62].

Zhang et al. (2022) investigated the prediction of MC of single-grain corn using technology manufactured by Specim Company of Finland in combination with deep learning [63]. All the analyses were performed in Python 3.7.5 software. The combination of the CNN-LSTM model offered high accuracy in predicting corn moisture content based on the HSI technology.

An HSI technology (Specim FX17, Spectral Imaging Ltd., Oulu, Finland) in the spectral range of 900–1700 nm was adopted for rapid and non-destructive determination of MC, free fatty acids (FFA) and peroxide value (PV) of the shelled almonds [28]. All analyses were performed in MATLAB 2019b software. Among the five pre-processing methods, the results showed that the SG-1st derivative and SG-SNV algorithms are the best pre-processing methods, while the PLSR model was recognised as the best regression model to detect the moisture content of shelled almonds with high accuracy.

The moisture content of peanuts was assessed using a Finnish HSI technology manufactured by Oulu company [64] according to several steps (Figure 5). First, data were evaluated by Unscrambler X 10.5 and MatlabR2013a software using the PLSR regression model. Twenty wavelengths were reported as the best wavelengths for predicting MC.

The HSI technology of the IMEC model (IMEC Kapeldreef 75, 3001 Heverlee, Belgium) was adopted using three pre-processing methods as implemented in Unscrambler software (10.5, CAMO, Trondheim, Norway) to predict and determine the moisture content of blueberry during drying. The regression coefficients were high at 706, 790, 827, 868 and 894 nm, and these wavelengths were introduced as the best wavelength for the prediction of MC. The two pre-processing methods of SVN and MSC, along with the PLS model, were the best models for MC prediction [34]. The researchers recommended using hyperspectral imaging techniques to predict the MC of blueberries during drying.

In another study, researchers determined MC of the melon samples during drying by the hot air method (temperature 60 °C and air speed of 2.5 m/s) with various pre-treatments, including ultrasonic (US), vacuum (VC) and US+VC pre-treatments, in different time intervals (0, 15, 30, 45, 60, 90, 120, 150, 180 and 210 min) by the HSI technology [5]. They performed spectral measurements with an HSI camera made in Finland (Specim, SisuChema—Finland) in the spectral range of 1000–2500 nm. The data were analysed by different pre-processing methods, such as standard normal variation—SNV, derived by filters Savitzky-Golay, multiplicative scatter correction—MSC and smoothing, while the PLS model was employed to evaluate the moisture content of melon. All analyses were performed in Hypertools and MATLAB^®^ 2010a software. The results showed promising hyperspectral images in evaluating the drying process of the melon.

**Table 2 molecules-28-02930-t002:** Application of HSI for prediction of MC of agricultural products during the drying process.

Products	Drying Method	Spectral Range (nm)	Data Analysis Method	Best Wavelength	Performance	Reference
Fresh carrot, celery stem, potato, and spinach leaves	Microwave-vacuum	950–1655	PLSR SVM MLR	1190 and 1450	R^2^_p_ = 0.974 RMSE_P_ = 4.70%	[3]
Apple slices	Hot air	396–1010	PLS	580, 750, 970	R^2^_p_ = 0.98 RMSE_P_ = 0.27	[59]
Apple slices	Convection	400–1000	PLSR	540, 817, 977	R^2^_p_ = 0.99 RMSE_P_ = 0.13	[4]
Apple slices	Hot-air	400–1700	PLSR	980 and 1450	R^2^_p_ = 0.99 RMSE_P_ = 0.89	[11]
Persimmon	In the shade	400–900	PLSR PCR LS-SVR RBFNN	about twenty wavelengths	R^2^_p_ = 0.856 RMSE_P_ = 0.0387	[8]
Wolfberry	Hot air	400–1001	SVM ABC GWO	895.28	R^2^_p_ = 0.9666	[60]
Potato	Oven drying	387–1035	XGBoost	400	R^2^_p_ = 0.8908 RMSE_P_ = 0.0610	[61]
Carrot	Hot air	400–1010	PLSR	973	R^2^_p_ = 0.90 RMSE_P_ = 0.0816	[62]
Corn	Oven	968.05–2575.05	CNN-LSTM	---	R^2^_p_ = 0.947 RMSE_P_ = 0.274	[63]
Almonds	Oven	900–1700	PLSR	970, 1001, 1154, 1312, 1350, 1437, 1670	R^2^_p_ = 0.941 RMSE_P_ = 0.494	[28]
Peanut	Oven	900–1700	PLSR	Twenty wavelengths	R^2^_p_ = 0.9445 RMSE_P_ = 1.9519	[64]
Blueberry	Climatic chamber	470–900	PLS	706, 790, 827, 868, and 894	R^2^_p_ = 0.9445 RMSE_P_ = 1.9519	[34]
Melon	Hot air	1000–2500	PLS	1400 and 1900	R^2^_p_ = 0.98 RMSE_P_ = 2.98	[5]

R^2^_p_: the root mean square error in prediction determination coefficients of prediction and RMSE_P_: the root mean square error of prediction.

### 6.2. The Possibility of Using Hyperspectral Imaging for a Smart System

In recent years, the development of suitable methods and technologies for continuous observation of the drying process and, thus, process control has received much attention. Smart drying is one of the emerging drying technologies within a multidisciplinary and interdisciplinary field. Its recent developments include the following research and development areas: artificial intelligence, biomimetics, computer vision and microwave/dielectric spectroscopy, visible and infrared spectroscopy, near-infrared (NIR), cloud/multispectral imaging and magnetic resonance imaging, ultrasound imaging, electrostatic sensing and control system for the drying environment [65].

As previously mentioned, hyperspectral imaging (HSI) includes the simultaneous spatial and spectral detection of the product non-invasively in a wide range of wavelengths. It is referred to as a versatile technology. Therefore, by modelling the data obtained through the hyperspectral image during drying, useful information about the chemical substances and physicochemical changes of the product can be obtained. With all this information, a smart dryer can be designed. The researcher’s results showed that despite the necessity to overcome the significant computational effort required for data processing, a significant step could be taken towards the HSI development on a large scale as a rapid and non-destructive detection method based on an intelligent dryer system. However, there are many limitations to the transfer of HSI technology from the laboratory to the industrial (real-time system) besides on many advantages of HSI. These restrictions include the high cost of HSI instruments and a lot of redundant data, which restrict the online processing of hyperspectral data. However, there are some research trends for reducing these limitations in the future. Firstly, cost reduction of HSI instruments must be attempted to develop and improve the hardware and software of the HSI system. Second, the existence of a lot of redundant data will be resolved with further advances in computational technology. So, despite the necessity to overcome these limitations, a significant step can be taken towards developing large-scale HSI-based intelligent dryer systems.

## 7. Conclusions and Future Trends

In this review, the feasibility of the HSI technology in predicting and measuring MC of different agricultural products during drying was explored by reviewing previous works. The results showed the applicability of non-invasive methods such as HSI to measure the MC of different agricultural products during drying. Some researchers have recently proposed machine learning to analyse spectral data and introduced the HSI technology combined with machine learning as a promising tool for the non-destructive and accurate detection of MC during drying. Future research can focus on the application of the HSI technology during the drying process. By expanding the studies on the integration of the HSI technology with machine learning, the prediction accuracy of MC can be increased and the chemical composition can be controlled during the drying process to take major steps towards developing a smart dryer as quickly as possible. In other words, this review suggests developing a machine learning system to create a smart dryer using the HSI analysis for future research. Therefore, by having smart dryers, high-quality and low-price products can be offered to consumers all over the world.

## Figures and Tables

**Figure 1 molecules-28-02930-f001:**
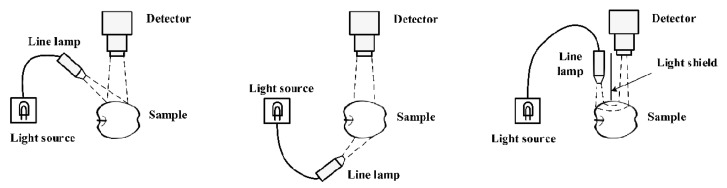
Various modes of HSI technology [26].

**Figure 2 molecules-28-02930-f002:**
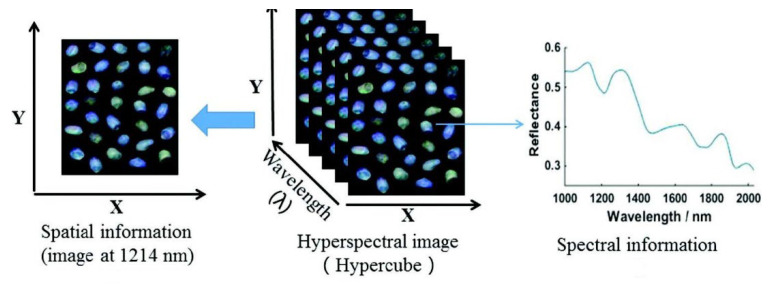
Spectrum and spatial information of peanut samples obtained by the HSI technology [29].

**Figure 3 molecules-28-02930-f003:**
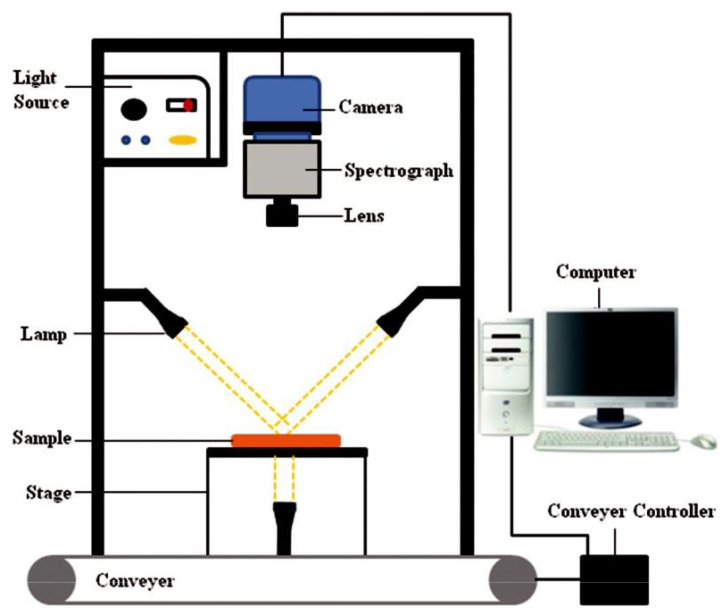
The main components of the HSI system [29].

**Figure 4 molecules-28-02930-f004:**
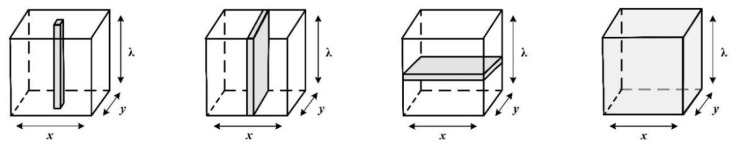
HSI acquisition method [26].

**Figure 5 molecules-28-02930-f005:**
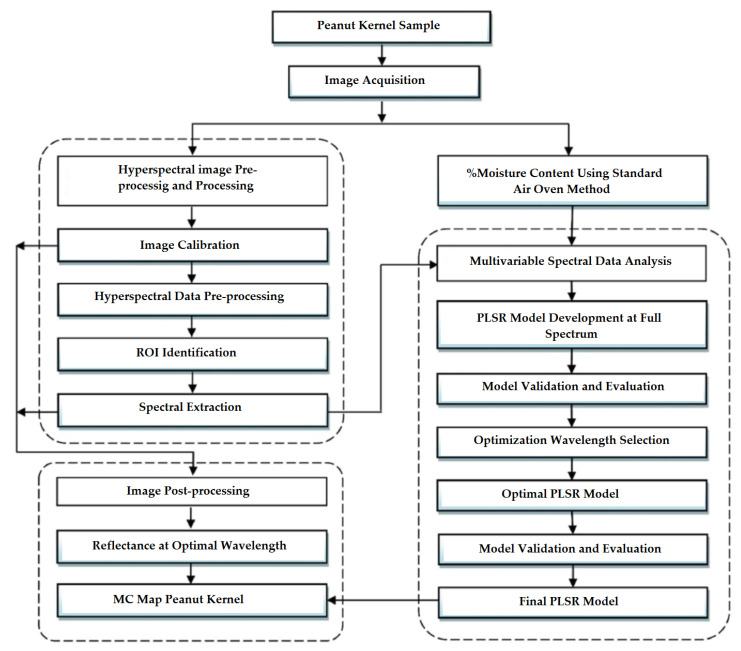
General steps involved in the experiment.

**Table 1 molecules-28-02930-t001:** The moisture content recommended for different crops during drying.

	Drying Method	Drying Time (min)	Initial Moisture Contents (%)	Final Moisture Contents (%)	Time Intervals (min)	Reference
Carrot slices	Microwave-vacuum	50	87	11 ± 2	0, 6, 12, 16, 20, 24.5, 29, 35, 41, 50	[3]
Celery stalks	Microwave-vacuum	53	95	16 ± 6	0, 15, 20, 24.5, 29, 31.5, 38, 42.5, 47, 53	[3]
Potato slices	Microwave-vacuum	30.5	85	12 ± 2	0, 3, 5.5, 8, 12.5, 17, 21.5, 24.5, 27.5, 30.5	[3]
Spinach leaves	Microwave-vacuum	20	93	8 ± 3	0, 4, 6, 8, 10, 12, 14, 16, 18, 20	[3]
Apple	Convection	240	88	16 ± 2	0, 30, 60, 90, 120, 150, 180, 240	[4]
Melon	Hot air	90	40	15	10, 15, 30, 50, 90	[5]
ginger slices	Microwave -vacuum	80	66.2	14.1	0, 25, 40, 55, 80	[6]

## Data Availability

All data generated or analysed during this study are included in the article. All authors read and approved the final manuscript.

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
