# Peer review of "The Application of Hyperspectral Imaging Technologies for the Prediction and Measurement of the Moisture Content of Various Agricultural Crops during the Drying Process"

_molecules, 2023, doi:10.3390/molecules28072930_

Round 1

Reviewer 1 Report

1.      The manuscript needs grammar improvement.

2.      This manuscript focused on reviewing hyperspectral imaging is not thoroughly done, and no new inferences were made. What is discussed in this work has been reported severally in the literature. See examples below.

-        https://www.sciencedirect.com/topics/medicine-and-dentistry/hyperspectral-imaging

-        https://www.mdpi.com/2072-4292/12/16/2659

-        https://www.sciencedirect.com/science/article/pii/B9780123747532100012

-        https://www.tandfonline.com/doi/full/10.1080/05704928.2013.838678?casa_token=AEDvIUhB6E8AAAAA%3AcCQd39kDyDyePDRvOSMs91aMpUfbonqgMj2z1LXADbE5q9TsLpf3UPWdaDk2c3JqvYX6PwFIdhhjYA

-        https://www.sciencedirect.com/science/article/pii/B9780123747532100024

3.      The authors should be thorough and focus on what they are contributing, as it is not clear in the current version.

Author Response

Dear editor,

We would like to express our sincere and deep thanks to the reviewers for the careful reading of the manuscript and for valuable suggestions to improve the quality of our manuscript entitled The application of the hyperspectral imaging technologies to predict and measure the moisture content of various agricul-tural crops during the drying process with manuscript ID of “molecules-2228049”. Responses to the associate reviewer’s comments were made in regard to the following his/her remarks in the revised version of the paper.

If you have any further queries, please do not hesitate to contact me.

Thank you for your consideration of this manuscript.

Kind regards,

Prof. Dr. Ebrahim Taghinezhad

Postdoctoral Fellowship

Wroclaw University of Environmental and Life Sciences

Department of Food Chemistry and Biocatalysis

Norwida 25, 50-375 Wroclaw, Poland

Mobile phone: 0048 600 651 226

Reply to respect reviewers:

Reviewer 1:

1.The manuscript needs grammar improvement.

Answer: It was done by track change.

2.This manuscript focused on reviewing hyperspectral imaging is not thoroughly done, and no new inferences were made. What is discussed in this work has been reported severally in the literature. See examples below.

-        https://www.sciencedirect.com/topics/medicine-and-dentistry/hyperspectral-imaging

-        https://www.mdpi.com/2072-4292/12/16/2659

-        https://www.sciencedirect.com/science/article/pii/B9780123747532100012

-        https://www.tandfonline.com/doi/full/10.1080/05704928.2013.838678?casa_token=AEDvIUhB6E8AAAAA%3AcCQd39kDyDyePDRvOSMs91aMpUfbonqgMj2z1LXADbE5q9TsLpf3UPWdaDk2c3JqvYX6PwFIdhhjYA

-        https://www.sciencedirect.com/science/article/pii/B9780123747532100024

 Answer: Thanks a lot for the best suggestion to complete this manuscript. We have completed the manuscript by these references in following:

Line 104 to 118: “The advantages of hyperspectral imaging over the traditional methods include minimal sample preparation, non-destructive nature, fast acquisition times and visualising spatial distribution of numerous chemical compo-sitions simultaneously. In recent years, the hyperspectral imaging technique has been regarded as a smart and promising analytical tool for research, control and industries [11]. The HSI systems are used in a wide range of applications such as agriculture, mineralogy, surveillance and target identification, astronomy, chemical imag-ing, environmental studies and the medical field [9]. The application of HSI in agriculture includes the evalua-tion of the nutrient status (e.g. nitrogen deficiency), monitoring crop disease, estimation of crop biochemical and biophysical characteristics of the crop (e.g. carotenoids, chlorophyll and water contents, LAI, biomass) to under-stand the physiological status of the vegetation and the prediction of yield and investigation of the properties of the soil (e.g. moisture, organic matter and carbon of the soil) [10]. The literature reports food-related studies in which hyperspectral technology was also applied for the detection of fungal contamination, bruising in apples, faecal contamination, skin tumours in chicken carcasses, grain inspections and so on [12]. The use of HSI tech-nology for the measurement of the MC of agricultural products during the drying process has been extensively explored [13-17]. The need for fast and careful methods has lead to the application of the HSI technology for MC control during the drying process. Moreover, the development of HSI instruments, a new method for data pro-cessing, will allow this technology to dominate in the future”.

Line 561-587: “Different techniques are available to determine moisture content during the drying of food such as com-puter vision technology (for ginseng roots [39], apple [40]), Near Infrared Reflectance Spectroscopy (for fer-mented sausages [41], apple [42]), MRI (for wheat [43], ham [44]), microwave dielectric spectroscopy (for ki-wifruit [45], pork meat [46]). There are many publications for the prediction of MC during various drying methods (e.g. for mango [47], blueberry [48] and apple slices [49]). More evaluation of moisture content by NIRS can be found in [50,51]. These techniques have some disadvantages that need to be considered. The computer vi-sion method is not usable for large-scale drying processes in which opaque metal chambers are used. Therefore, improving the scope of use would be a chance for the development of this technology. NIRs method needs cali-bration based on reference techniques. In addition, the cost of equipment is relatively high and multiple sensors must be installed to monitor local moisture content or particle size in the dryer. Moreover, this technique is in-sensitive to impurities, and only the surface moisture of the material can be determined due to the short wave-length. The use of MRI-enabled visualisation of spatial changes during drying also proves that the MRI method offers very valuable information for process optimisation in the commercial processing of different varieties of food materials, but its application for the online system has a high cost. Microwave dielectric spectroscopy needs more expensive electronic components due to the requirement of applying high frequency, and it must be cali-brated separately for different materials. It is difficult or even infeasible to attain an appropriate spatial resolu-tion because of the relatively long wavelengths. In conclusion, the commercial application of microwave dielec-tric spectroscopy still needs further research [52]. Visible and near-infrared (VIS/NIR) spectroscopy was report-ed as one of the most rapid and appropriate techniques for the prediction and detection of different parameters such as rice and its flour quality [53], tomato pesticide residues [54,55], winter wheat leaf [56] beside the predic-tion of MC. But the most important advantages of the application of the HSI technology in the food industry ra-ther than other techniques are as follows [11]:

No requirement for sample preparation, environmentally friendly, non-invasive, and non-destructive method,  economical as compared with traditional techniques, recording a spectral volume as compared to spectroscopy (collecting a single spectrum at one spot on a sample), obtaining both qualitative and quantitative properties, reporting the spatial distribution and concentration of the chemical composition of samples, distin-guishing samples with similar colours, incomparable for process monitoring and real-time inspection”

 3.The authors should be thorough and focus on what they are contributing, as it is not clear in the current version.

Answer: Our team research published the following references which added to main text for section 6 in line 579-581: “Visible and near-infrared (VIS/NIR) spectroscopy was reported as one of the most rapid and appropriate tech-niques for the prediction and detection of different parameters such as rice and its flour quality [53], tomato pesticide residues [54,55], winter wheat leaf [56] beside the prediction of MC.”.  Also, we are performing a project for the prediction of the moisture content of parboiled paddy during drying by hyperspectral imaging in Poland. This project was funded by the NAWA – Polish National Agency for Academic Exchange under the Ulam NAWA Programme (Project No. BPN/ULM/2021/1/00231). So, before starting the project, we studied the different related papers and we understood there is not any review paper only for the prediction of MC of agricultural different products during drying.

Reviewer 2 Report

The authors still need to address important concerns:

-       Why is the moisture content important for agricultural crops during drying? This needs to be addressed through a table that states the moisture content recommended for different crops either during drying.

-       I would like to see a section where hyperspectral imaging is compared with other sensors like Visible/NIR spectroscopy, and other moister content sensing technologies and what are the advantage for hyperspectral imaging over other sensors.

-       The manuscript needs to include the possibility of using hyperspectral imaging for digital food manufacturing (in this case, drying) and how hyperspectral imaging can be deployed for such a purpose.     

Author Response

Dear editor,

We would like to express our sincere and deep thanks to the reviewers for the careful reading of the manuscript and for valuable suggestions to improve the quality of our manuscript entitled The application of the hyperspectral imaging technologies to predict and measure the moisture content of various agricultural crops during the drying process with manuscript ID of “molecules-2228049”. Responses to the associate reviewer’s comments were made in regard to the following his/her remarks in the revised version of the paper.

If you have any further queries, please do not hesitate to contact me.

Thank you for your consideration of this manuscript.

Kind regards,

Prof. Dr. Ebrahim Taghinezhad

Postdoctoral Fellowship

Wroclaw University of Environmental and Life Sciences

Department of Food Chemistry and Biocatalysis

Norwida 25, 50-375 Wroclaw, Poland

Mobile phone: 0048 600 651 226

Reviewer 2:

The authors still need to address important concerns:

      1- Why is the moisture content important for agricultural crops during drying?

Answer: The flowing phrase was added in the introduction section:

“The decreasing quality due to over-drying of agricultural products can be pointed to as one of the most important problems during drying. So, the measurement of MC and the change in quality of agricultural products during the drying process is very necessary as those can affect the drying rate and energy consumption. In other words, the over drying and the rise in the temperature of the sample during the drying process can adversely af-fect the quality of the dried samples.”

      This needs to be addressed through a table that states the moisture content recommended for different crops either during drying.

     Answer: changing moisture content during drying have been shown in Table 1.

      2- I would like to see a section where hyperspectral imaging is compared with other sensors like Visible/NIR spectroscopy, and other moister content sensing technologies and what are the advantage of hyperspectral imaging over other sensors.

Answer: Thanks a lot for the best point. The following section was added to the manuscript:

 “ 6. Comparison of various techniques for prediction of moisture content of agricultural products

Different techniques are available to determine moisture content during the drying of food such as computer vision technology (for ginseng roots [39], apple [40]), Near Infrared Reflectance Spectroscopy (for fermented sausages [41], apple [42]), MRI (for the wheat [43], ham [44]), microwave dielectric spectroscopy (for kiwifruit [45], pork meat [46]). There are many publications for the prediction of MC during various drying methods (e.g. for mango [47], blueberry [48] and apple slices [49]). More evaluation of moisture content by NIRS can be found in [50,51]. These techniques have some disadvantages that need to be considered. The computer vi-sion method is not usable for large-scale drying processes in which opaque metal chambers are used. Therefore, improving the scope of use would be a chance for the development of this technology. NIRs method needs cali-bration based on reference techniques. In addition, the cost of equipment is relatively high and multiple sensors must be installed to monitor local moisture content or particle size in the dryer. Moreover, this technique is in-sensitive to impurities, and only the surface moisture of the material can be determined due to the short wave-length. The use of MRI-enabled visualisation of spatial changes during drying also proves that the MRI method offers very valuable information for process optimisation in the commercial processing of different varieties of food materials, but its application for the online system has a high cost. Microwave dielectric spectroscopy needs more expensive electronic components due to the requirement of applying high frequency, and it must be cali-brated separately for different materials. It is difficult or even infeasible to attain an appropriate spatial resolu-tion because of the relatively long wavelengths. In conclusion, the commercial application of microwave dielec-tric spectroscopy still needs further research [52]. Visible and near-infrared (VIS/NIR) spectroscopy was reported as one of the most rapid and appropriate techniques for the prediction and detection of different parameters such as rice and its flour quality [53], tomato pesticide residues [54,55], winter wheat leaf [56] beside the predic-tion of MC. But the most important advantages of the application of the HSI technology in the food industry ra-ther than other techniques are as follows [11]:

No requirement for sample preparation, environmentally friendly, non-invasive, and non-destructive method,  economical as compared with traditional techniques, recording a spectral volume as compared to spectroscopy (collecting a single spectrum at one spot on a sample), obtaining both qualitative and quantitative properties, reporting the spatial distribution and concentration of the chemical composition of samples, distin-guishing samples with similar colours, incomparable for process monitoring and real-time inspection.

     3- The manuscript needs to include the possibility of using hyperspectral imaging for digital food manufacturing (in this case, drying) and how hyperspectral imaging can be deployed for such a purpose.     

Answer: Thanks a lot for the best comment we could add the following section to the manuscript:

Line 894 to 920: “6.2. The possibility of using hyperspectral imaging for a smart system

In recent years, the development of suitable methods and technologies for continuous observation of the drying process and thus process control has received much attention. Smart drying is one of the emerging drying tech-nologies within a multidisciplinary and interdisciplinary field, and its recent developments include the follow-ing research and development areas: artificial intelligence, biomimetics, computer vision and micro-wave/dielectric spectroscopy, visible and infrared spectroscopy, near infrared (NIR), cloud/multispectral imag-ing and magnetic resonance imaging, ultrasound imaging, electrostatic sensing and control system for the dry-ing environment [65].

As previously mentioned, hyperspectral imaging (HSI) includes the simultaneous spatial and spectral detection of the product non-invasively in a wide range of wavelengths, and it is referred to as a versatile technology. Therefore, by modelling the data obtained through the hyperspectral image during drying, useful information about the chemical substances and physicochemical changes of the product can be obtained, and with all this information, a smart dryer can be designed. The results of the researchers showed that despite the necessity to overcome the significant computational effort required for data processing, a significant step can be taken to-wards the HSI development on a large scale as a rapid and non-destructive detection method based on an intel-ligent dryer system. There are many limitations to the transfer of the HSI technology from the laboratory to the industrial (real-time system) besides on many advantages of HSI. These restrictions include the high cost of HSI instruments and a lot of redundant data which restrict the online processing of hyperspectral data. There are some research trends for the reduction of these limitations in the future. Firstly, cost reduction of HSI instru-ments must be attempted to develop and improve the hardware and software of the HSI system. Second, the ex-istence of a lot of redundant data will be resolved with further advances in computational technology. So, de-spite the necessity to overcome these limitations, a significant step can be taken towards the development of large-scale HSI-based intelligent dryer systems.”

Round 2

Reviewer 1 Report

Please improve the grammar

Author Response

Dear Reviewer,

We would like to express our sincere and deep thanks for your careful reading of the manuscript and valuable suggestions to improve the quality of our manuscript entitled The application of the hyperspectral imaging technologies to predict and measure the moisture content of various agricultural crops during the drying process with the manuscript ID of “molecules-2228049”. Grammar has been improved by the Native Language Company in Poland.

If you have any further queries, please do not hesitate to contact me.

Thank you for your consideration of this manuscript.

Kind regards,

Prof. Dr. Ebrahim Taghinezhad

Postdoctoral Fellowship

Wroclaw University of Environmental and Life Sciences

Department of Food Chemistry and Biocatalysis

Norwida 25, 50-375 Wroclaw, Poland

Mobile phone: 0048 600 651 226
